# Relationship between the Surface Roughness of Material and Bone Cement: An Increased “Polished” Stem May Result in the Excessive Taper-Slip

**DOI:** 10.3390/ma14133702

**Published:** 2021-07-02

**Authors:** Masayuki Hirata, Kenichi Oe, Ayumi Kaneuji, Ryusuke Uozu, Kazuhiro Shintani, Takanori Saito

**Affiliations:** 1Department of Orthopaedic Surgery, Kansai Medical University, 2-5-1 Shinmachi, Hirakata, Osaka 573-1010, Japan; hiratamasayuki.08.11@gmail.com (M.H.); saitot@hirakata.kmu.ac.jp (T.S.); 2Department of Orthopaedic Surgery, Kanazawa Medical University, 1-1 Daigaku, Uchinada, Ishikawa 920-0293, Japan; kaneuji@kanazawa-med.ac.jp; 3Department of Mechanical Engineering, Kanazawa Institute of Technology, 3-1 Yatsukaho, Hakusan, Ishikawa 924-0838, Japan; b1608555@planet.kanazawa-it.ac.jp (R.U.); shintani@neptune.kanazawa-it.ac.jp (K.S.)

**Keywords:** surface roughness, surface wettability, frictional coefficient, bone cement, polished tapered stem, periprosthetic femoral fracture

## Abstract

Although some reports suggest that taper-slip cemented stems may be associated with a higher periprosthetic femoral fractures rate than composite-beam cemented stems, few studies have focused on the biomaterial effect of the polished material on the stem–cement interface. The purpose of this study was to investigate the relationship between surface roughness of materials and bone cement. Four types of metal discs—cobalt-chromium-molybdenum alloy (CoCr), stainless steel alloy 316 (SUS), and two titanium alloys (Ti-6Al-4V and Ti-15Mo-5Zr-3Al)—were prepared. Five discs of each material were produced with varying degrees of surface roughness. In order to evaluate surface wettability, the contact angle was measured using the sessile drop method. A pin was made using two bone cements and the frictional coefficient was assessed with a pin-on-disc test. The contact angle of each metal increased with decreasing surface roughness and the surface wettability of metal decreased with higher degrees of polishing. With a surface roughness of Ra = 0.06 μm and moderate viscosity bone cement, the frictional coefficient was significantly lower in CoCr than in SUS (*p* = 0.0073). In CoCr, the low adhesion effect with low frictional coefficient may result in excessive taper-slip, especially with the use of moderate viscosity bone cement.

## 1. Introduction

The introduction of Charnley’s low friction arthroplasty was a technological revolution and has become the most successful total hip arthroplasty (THA) [1]. Charnley made three major contributions to the evolution of THA: (1) the concept of low friction torque arthroplasty, (2) the use of acrylic cement to fix components to living bone, and (3) the introduction of high-density polyethylene as a bearing material [2]. The excellent clinical long-term results of Charnley’s low friction arthroplasty have been reported in many studies [3,4], and the fundamental concepts of Charnley’s THA still exist today. The Exeter (Stryker, Mahwah, NJ, USA) is a new stem with a collarless tapered shape that was introduced because the existing collars could not regularly and reliably transmit load to the femoral neck. The taper-slip concept allows the stem to subside and become lodged as a wedge in the cement mantle during loading [5]. In vitro, polished stems have been shown to provide greater potential to develop “taper-lock” fixation [6,7] and stem subsidence in polished stems results in compressive force on cement and cement creep [8]. Ling et al. reported 94% survival of the original Exeter stem at a 35 years follow up, with aseptic loosening as the endpoint [9]. Currently, “collarless, tapered, polished” stems have become widely used and they are considered the “golden standard” of cemented stems. In short, the design of cemented stems involves two concepts: a composite-beam or shape-closed design originated by Charnley and a taper-slip or force-closed design rooted in Exeter [10].

Periprosthetic femoral fractures (PPFs) following THA are devastating complications that are associated with functional limitations and increased overall mortality [11]. The Nordic Arthroplasty Register Association database reported a 7-fold increase in the incidence of PPF with uncemented stems compared to cemented stems [12], despite the fact that use of uncemented fixation in THA is increasing worldwide [13]. However, with cemented stems, some authors have reported that the risk of PPF for taper-slip stems was high compared to composite-beam stems and that there was a difference among taper-slip stems [12,14,15,16]. Using the Exeter or CPT (Zimmer, Warsaw, IN, USA) stem, Grammatopoulos et al. demonstrated the different fracture patterns among well-fixed cemented taper-slip stems and highlighted a specific fracture pattern with both splitting and spiral elements [17]. Biomechanically, the C-stem (DePuy International, Leeds, UK) and the CPT stem created a different fracture pattern under the same loading condition [18]. These results enable speculation that the excessive taper-slip possibly causes PPF, although experimentally, the taper-slip stem can support substantially greater loads before failure than the composite beam [19]. Furthermore, the effect of the surface appearance on the stem–cement interface may be different for each metal because the subsidence is different for each taper-slip stem and the metal implant surface appearance is only defined by human beings [20,21]. The question “what is polished?” is ultimately posed and there have been a few reports that have specifically assessed the effect of the polished material on the stem–cement interface. The aim of this study was to evaluate the relationship between the surface roughness of metallic material and bone cement. We hypothesised that the effect of the surface appearance on the stem–cement interface will be different for each metal, even if the roughness of the implant surface is equal.

## 2. Materials and Methods

### 2.1. Materials

Four types of metal discs—cobalt-chromium-molybdenum alloy (CoCr), stainless steel alloy 316 (SUS), and two titanium alloys (Ti-6Al-4V and Ti-15Mo-5Zr-3Al)—were prepared (Table 1). The shape of each metal disc was 45 mm in diameter and 5 mm in thickness and five discs of each material were produced. Each metal disc was polished with #2000, #500, and #120 waterproof paper; glass beads and an alumina shot blast; and adjusted to varying degrees of roughness. The disc surface roughness was randomly measured at five points on each disc by the instrument (Surfcom E-ST-S03A; Tokyo Seimitsu Co., Tokyo, Japan) and was determined as an average. The conditions were traversing length free, cut-off 0.25 mm and range 0.1–5 (Table 2). In order to demonstrate the texture, scanning electron microscope images of the disc surfaces were assessed using a digital microscope (VHX-6000; KEYENCE Co., Osaka, Japan) (Figure 1). In addition, profilometer images of the disc surfaces were evaluated using a laser scanning wide-area 3D measurement system (VR-5000; KEYENCE Co., Osaka, Japan) (Figure 2).

### 2.2. Experiment 1: Evaluation of Surface Wettability

Surface wettability is one of the characteristics of a metal surface and indicates whether the metal has a water-shedding or water-attracting surface [22,23]. For the evaluation of surface wettability, a contact angle was used. A low contact angle means the surface is water-attracting (hydrophilic) and a high contact angle means the surface is water-shedding (hydrophobic).

The contact angle measurements were carried out using the sessile drop method with an optical contact angles meter (FTA 1000; JUSCO International Co., Tokyo, Japan). Saline was drawn into a micrometric syringe and dropped onto a metal disc (1.0 μL). After 28 s, the droplet was captured using a camera (ArtCAM USB2.0 Camera; ARTRAY, Tokyo, Japan). Temperature (22–23 °C) and relative humidity (50–55%) were maintained during all analyses. Then, the captured image was analysed using FTA 32 software (Version 1, JUSCO International Co., Tokyo, Japan) (Figure 3). The contact angle was measured by examining the angle formed between the solid and the tangent to the drop surface. Three separate measurements were taken and the results were averaged.

### 2.3. Experiment 2: Evaluation of Frictional Coefficient

For the evaluation of a frictional coefficient, a pin-on-disc test was performed. At first, a pin was made by two bone cements (Endurance (DePuy CMW, Blackpool, UK) and Palacos R (Heraeus Kulzer, Wehrheim, Germany)) using vacuum mixing. Endurance, in which the viscosity is similar to that of Smart Set MV (DePuy CMW, Blackpool, UK), represented a moderate viscosity bone cement, whereas Palacos R represented a high viscosity bone cement. After the hardening of the bone cement, the pin was polished using a lathe and #400 sandpaper. The pin was attached to the pin-on-disc tribometer (FPR-2100; RHESCA Co., Ltd., Tokyo, Japan) and the contact surface was polished further using #800–#2000 sandpapers and #2000 emery paper, with 20 rpm of 1N. The shape of the pin was 5 mm in diameter and 45 mm in length (Figure 4).

The tribological experiments were performed using the pin-on-disc tribometer (RHESCA Co., Ltd., Tokyo, Japan), according to the report by Fukui et al. [24]. The disc specimen was put on the plate with circulated saline (37 °C) and sliding friction was applied by rotating the prepared bone cement pin against the disc surface. The pin made by Endurance was set on the disc with a radius of 4.0 mm, under a load weight of 500 g, and the disc was rotated at a speed of 0.050 mm/s, with 0.1 rpm for 30 s. The pin made by Palacos R was set on the disc with a radius of 10.0 mm, under a load weight of 500 g. In order to standardise the conditions, the disc was rotated at a speed of 0.105 mm/s, with 0.1 rpm for 30 s (Figure 5). The dynamic frictional coefficient was calculated as the acquired frictional force divided by the weight (500 g) by using the FRP Data Analysis System (version 1.11.2; RHESCA Co., Ltd., Tokyo, Japan). Five separate measurements were taken and the results averaged.

### 2.4. Statistical Analysis

Using a generalised linear mixed effects model with surface roughness as a random effect, a quadratic curve was drawn using plots between the surface roughness and frictional coefficient in each metal. Then, the corrected values of the surface roughness were calculated with adjustments within 0.03 and the actual measured or corrected values were identified for comparison with even surface roughness. In order to compare the surface roughness between groups, a generalised linear mixed effects model with surface roughness was employed again. Statistical significance was set at *p* < 0.05. All analyses were performed using SAS (Version 9.2, SAS Institute Inc., Cary, NC, USA).

## 3. Results

### 3.1. Experiment 1

The relationship between the contact angle and surface roughness is shown in Figure 6. The contact angle of each metal decreased with increasing surface roughness. Using a power approximation fitted curve, the expected contact angle with a surface roughness of Ra = 0.06 μm was 99.5, 93.0, 112.0, and 112.0 in CoCr, SUS, Ti-6Al-4V, and Ti-15Mo-5Zr-3Al, respectively. The contact angle with a surface roughness of Ra = 0.1 μm was 95.5, 89.5, 102.5, and 102.5 in CoCr, SUS, Ti-6Al-4V, and Ti-15Mo-5Zr-3Al, respectively. The contact angle with a surface roughness of Ra = 0.4 μm was 86.5, 81.0, 82.0, and 81.5 in CoCr, SUS, Ti-6Al-4V, and Ti-15Mo-5Zr-3Al, respectively. The contact angle of SUS was less than that of CoCr.

### 3.2. Experiment 2

The relationship between the frictional coefficient and surface roughness is shown in Figure 7. Using a power approximation fitted curve in Figure 7, the expected frictional coefficient is calculated. With a surface roughness of arbitrary Ra = 0.06 or 0.1 μm, the frictional coefficient of CoCr was less than that of SUS in both Endurance and Palacos R (Figure 8).

The comparison of frictional coefficients in the metal samples is shown in Table 3 and Table 4. By using Endurance with a surface roughness of Ra = 0.06 μm, the frictional coefficient in CoCr was significantly lower than that in SUS (*p* = 0.0073). By using Endurance and Palacos R with a surface roughness of Ra = 0.1 μm, the frictional coefficients in Ti-6Al-4V and Ti-15Mo-5Zr-3Al were significantly lower than that in CoCr (*p* < 0.0001). Using Endurance and Palacos R with a surface roughness of Ra = 0.4 μm, the frictional coefficient in Ti-15Mo-5Zr-3Al was significantly higher than in Ti-6Al-4V (*p* < 0.0001). With a surface roughness of Ra = 0.4 μm, the frictional coefficient in Ti-6Al-4V was significantly lower than in CoCr and SUS, whereas there was no significant difference between Ti-15Mo-5Zr-3Al and CoCr and SUS.

## 4. Discussion

Types of cemented stem fixation are classified as composite-beam design and taper-slip design [10]. Implants relying on a composite beam principle typically have a “satin” or “matte” surface finish to maximise the mechanical strength of the bond between the cement mantle and the stem. The taper-slip concept uses a dual-tapered or triple-tapered stem geometry typically with a “smooth” or “polished” surface finish allowing the implant to wedge into the cement mantle [25]. However, the effect of the surface appearance on the stem–cement interface may be different for each metal because the extent of subsidence is extremely sensitive to interface friction [26]. For the taper-slip design, the adequacy and safety of the “polished” surface finish of each metal remains unknown, with the average surface roughness (Ra, μm) of “polish” ranging from 0 to 0.1 (Table 5). Therefore, we assessed the effect of metallic surface roughness on the bone cement.

Commonly, there are three types of materials that are used for cemented stems—CoCr alloy, SUS, and titanium alloy. In polished stems, stems made of titanium alloy are not favoured because they are associated with early failure and two disadvantages: a stiffness of about 50% of that of CoCr alloy or SUS and a susceptibility of the material to crevice corrosion [27]. Conversely, CoCr alloy has been considered to behave similarly to SUS because they have a similar Young’s modulus and Poisson coefficient. However, Borruto et al. demonstrated that the surface wettability of the material has an effect on friction factors and the different surface wettability causes a different distribution of the drops of water [22]. Indeed, in this study, the hydrophobic nature of each metal increased with decreasing surface roughness because the contact angle of each metal increased with decreasing surface roughness. This means that the surface wettability of a metal is decreased in accordance with greater polishing. With a surface roughness of Ra = 0.06 μm, the contact angle of CoCr was larger than that of SUS. Consequently, CoCr had a lower surface wettability than SUS and tended not to adhere to bone cement. Furthermore, with a surface roughness of Ra = 0.06 μm, the frictional coefficient of CoCr was lower than that of SUS, particularly when using Endurance. These results indicate that, in CoCr, the low adhesion effect with low frictional coefficient may result in the excessive taper-slip. Using imitated stems made of CoCr (Ra < 0.02 μm) and SUS (Ra < 0.02 μm), Tsuda also demonstrated that the subsidence into the bone cement was significantly greater in CoCr than SUS [28]. He concluded that CoCr alloy and SUS alloy showed different mechanical behaviour within the bone cement and this difference can be caused by a difference in surface wettability between the two materials. In addition, Nelissen et al. reported that at the 2 year follow-up, the subsidence of the Exeter stem using low viscosity bone cement was greater than with high viscosity bone cement [29]. Although there were no differences in clinical revision rates based on viscosity [30], bone cement viscosity affected the frictional coefficient of the surface roughness in the current study. Excessively “polished” CoCr may not be compatible with the stem–cement interface using moderate viscosity bone cement.

Clinically, there is a difference among taper-slip stems in terms of the incidence of PPF. According to the National Joint Registry of England, Wales, and Northern Ireland database on 257,202 THAs, analysis of the four cemented femoral stem brands showed a PPF incidence of 0.12% in Exeter V40, 0.07% in the Charnley stem (DePuy International, Leeds, UK), 0.46% in the CPT stem, and 0.14% in the C-stem [16]. The incidence was lowest with the composite-beam stem (Charnley stem) and, of note, the PPF incidence with a CPT stem was the highest. Other authors have also reported that CPT stems are associated with a higher incidence of PPF [15,31]. The list of the major available cemented stems is shown in Table 6 and, of the taper-slip stems, only the CPT stem is made of CoCr. Although the difference in stem design may be involved in the higher incidence of PPF, the stem material or surface finish may also be one of the causes of PPF. A polish-tapered stem made of CoCr potentially results in PPF.

Some limitations of our study must be noted. First, the sample size was small. Although statistically significant data were acquired, the Ra parameters that could be compared were limited. Ideally, more samples should have been included. Second, the frictional coefficient between the metal and bone cement is dependent on the environment, that is, whether the condition is wet or dry [32]. In this study, however, only saline was used to simulate the body environment in both experiments. Furthermore, the surface finish of the pin made by bone cement could not be evaluated because the diameter was 5 mm and, therefore, too small. Third, this study was a biomaterial study, so results could differ in clinical practice. In order to confirm the relationship between the surface roughness of metallic material and bone cement, a biomechanical study, including a stem subsidence test and torsional stability test at a variety of degrees of surface roughness, should be considered.

## 5. Conclusions

The surface wettability of a metal decreases with increased polishing and CoCr had the lowest surface wettability compared to SUS with a surface roughness of Ra = 0.06 μm. Furthermore, the frictional coefficient of CoCr was lower compared to that of SUS. In CoCr, the low adhesion effect with low frictional coefficient may result in excessive taper-slip, especially when using moderate viscosity bone cement. However, further multifaceted studies that include biomechanical assessments are required to confirm our results.

## Figures and Tables

**Figure 1 materials-14-03702-f001:**
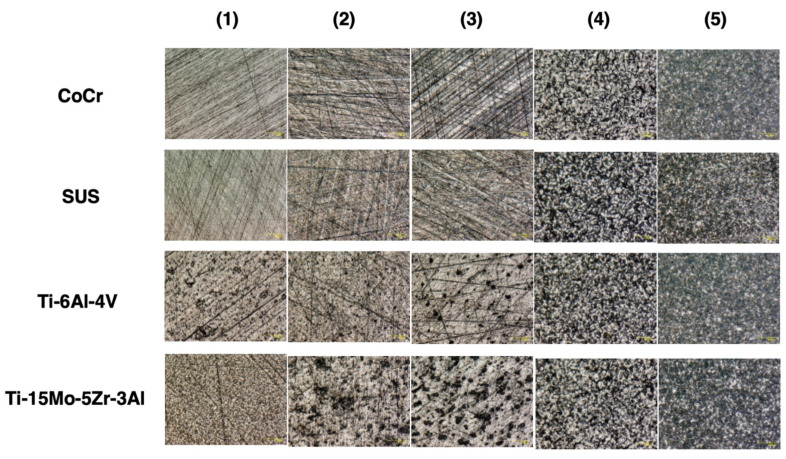
Photograph of the surface using a digital microscope (VHX-6000; KEYENCE Co., Osaka, Japan) (original magnification × 500).

**Figure 2 materials-14-03702-f002:**
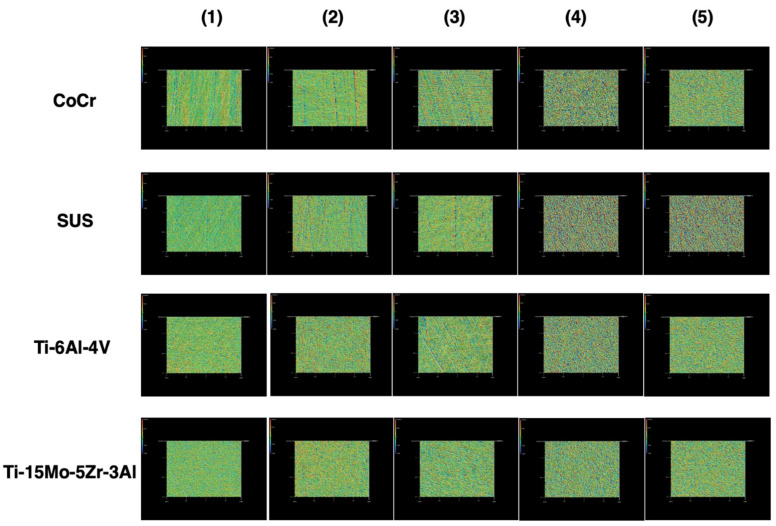
Photograph of the surface using a laser scanning wide-area 3D measurement system (VR-5000; KEYENCE Co., Osaka, Japan) (original magnification × 160).

**Figure 3 materials-14-03702-f003:**
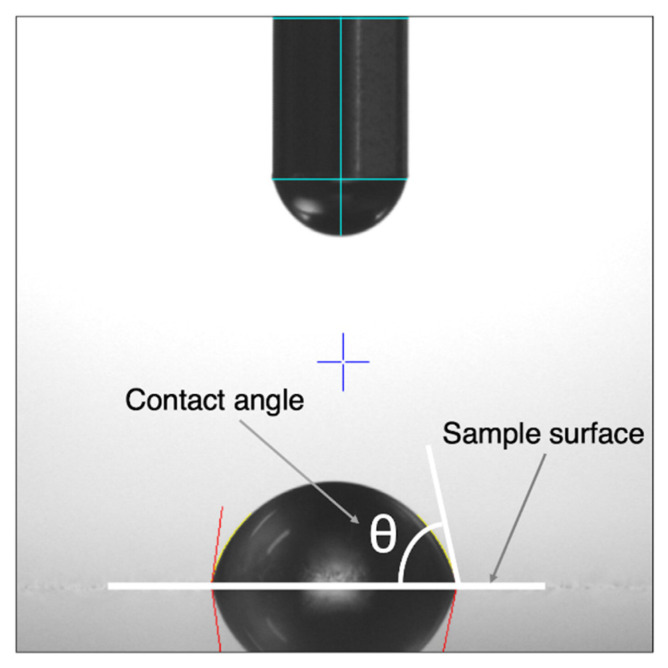
Photograph of the droplet on Ti-15Mo-5Zr-3Al (surface roughness Ra = 0.34, contact angle = 81.43°).

**Figure 4 materials-14-03702-f004:**
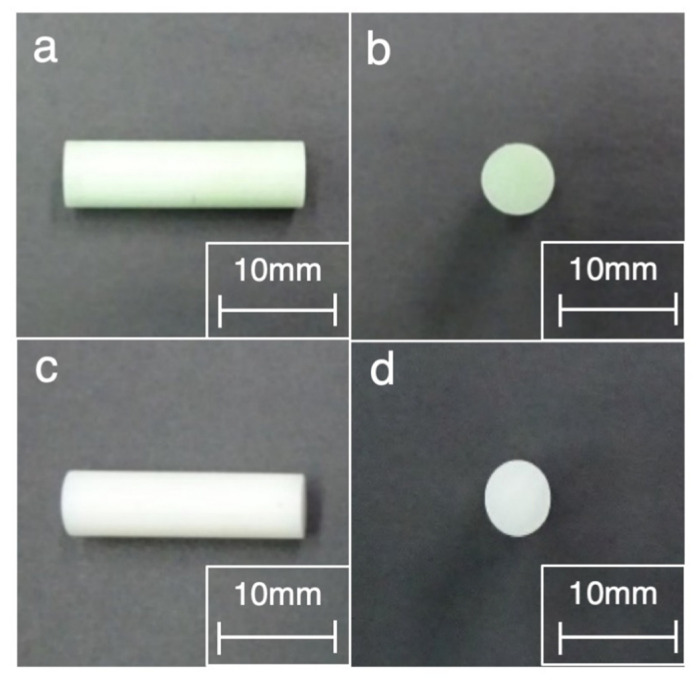
Photograph of the pin made by bone cement showing the following: (**a**) the side view of Endurance (DePuy CMW, Blackpool, UK), (**b**) the front view of Endurance, (**c**) the side view of Palacos R (Heraeus Kulzer, Wehrheim, Germany), and (**d**) The front view of Palacos R.

**Figure 5 materials-14-03702-f005:**
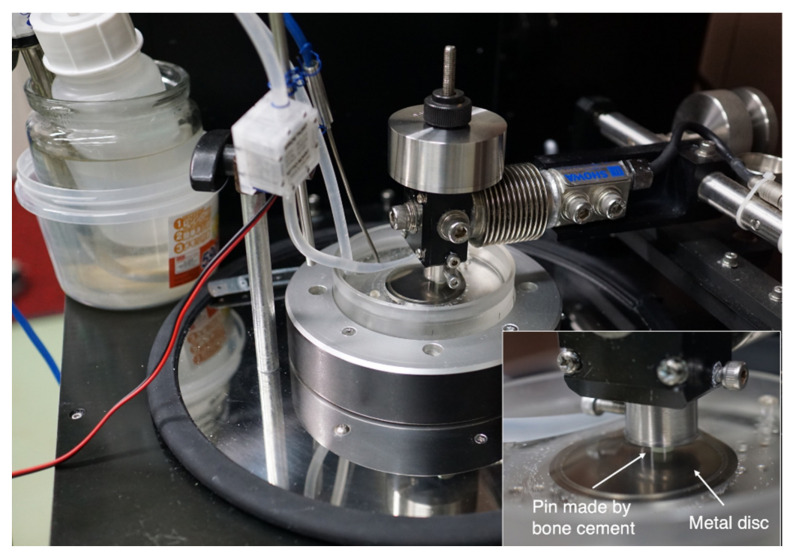
Photograph of the tribological experiments using a pin-on-disc tribometer (RHESCA Co., Ltd., Tokyo, Japan). The inset is a magnified view of the pin-on-disc tribometer.

**Figure 6 materials-14-03702-f006:**
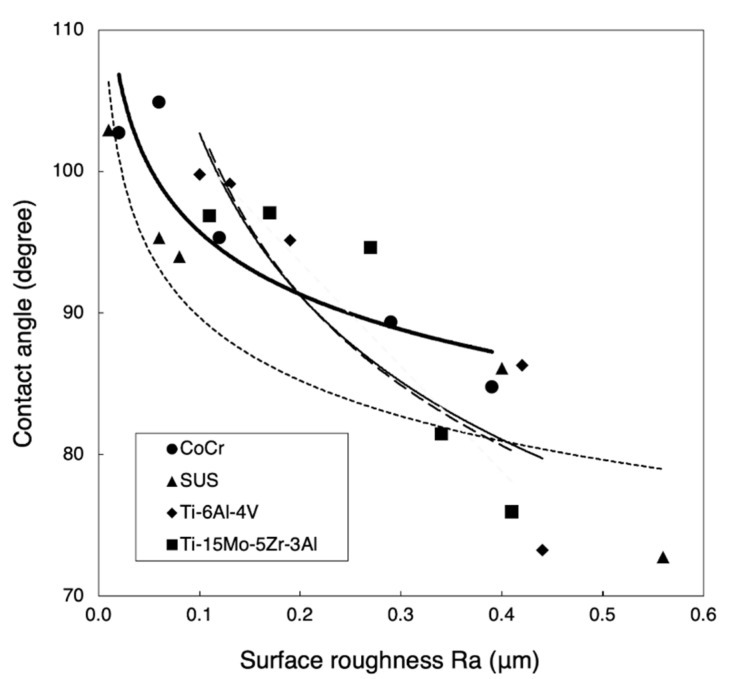
Relationship between the contact angle and surface roughness. The bold line, the dotted line, the solid line, and the dashed line indicate the power approximation fitted curve for CoCr, SUS, Ti-6Al-4V, and Ti-15Mo-5Zr-3Al.

**Figure 7 materials-14-03702-f007:**
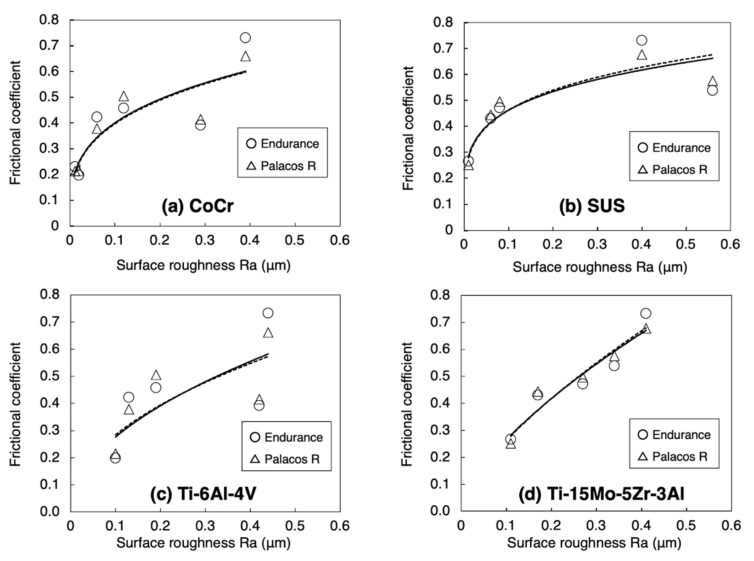
Relationship between the frictional coefficient and surface roughness: (**a**) CoCr, (**b**) SUS, (**c**) Ti-6Al-4V, and (**d**) Ti-15Mo-5Zr-3Al. The straight line and the dotted line indicate the power approximation fitted curve of Endurance (DePuy CMW, Blackpool, UK) and Palacos R (Heraeus Kulzer, Wehrheim, Germany).

**Figure 8 materials-14-03702-f008:**
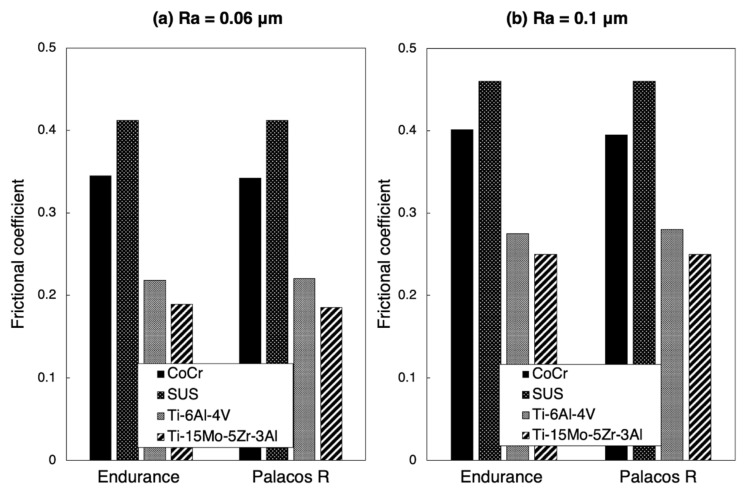
Frictional coefficient calculated by the power approximation fitted curve: (**a**) Ra = 0.06 μm and (**b**) Ra = 0.1 μm.

**Table 1 materials-14-03702-t001:** Chemical composition of each metal disc.

Material	Item (%)																
	Co	Cr	Mo	Fe	Ni	Ti	Al	V	Zr	Mn	Si	P	S	N	O	C	H
CoCr	65.1	27.8	5.6	0.1	0.04	(-)	(-)	(-)	(-)	0.7	0.6	(-)	(-)	0.1	(-)	0.05	(-)
SUS	0.2	16.9	2.0	68.2	10.2	(-)	(-)	(-)	(-)	1.3	0.5	0.3	0.3	(-)	(-)	0.05	(-)
Ti-6Al-4V	(-)	(-)	(-)	0.1	(-)	89.8	5.9	4.0	(-)	(-)	(-)	(-)	(-)	0.02	0.1	0.02	0.001
Ti-15Mo-5Zr-3Al	(-)	(-)	15.0	0.03	76.6	(-)	3.3	(-)	5.2	(-)	(-)	(-)	(-)	0.005	0.1	0.005	0.005

CoCr, cobalt-chromium-molybdenum alloy; SUS, stainless steel alloy; Co, Cobalt; Cr, Chromium; Mo, Molybdenum; Fe, Iron; Ni, Nickel; Ti, Titanium; Al, Aluminium; V, Vanadium; Zr, Zirconium; Mn, Manganese; Si, Silicon; P, Phosphorus; S, Sulfur; N, Nitrogen; O, Oxygen; C, Carbon; H, Hydrogen.

**Table 2 materials-14-03702-t002:** Surface roughness of each sample.

Material	Surface Roughness Ra (μm)				
	(1)	(2)	(3)	(4)	(5)
CoCr	0.02 ± 0.002	0.06 ± 0.002	0.12 ± 0.009	0.29 ± 0.014	0.39 ± 0.017
SUS	0.01 ± 0.002	0.06 ± 0.003	0.08 ± 0.005	0.40 ± 0.018	0.56 ± 0.030
Ti-6Al-4V	0.10 ± 0.004	0.13 ± 0.007	0.19 ± 0.019	0.42 ± 0.017	0.44 ± 0.027
Ti-15Mo-5Zr-3Al	0.11 ± 0.007	0.17 ± 0.023	0.27 ± 0.017	0.34 ± 0.009	0.41 ± 0.018

Data are presented as mean ± standard deviation. CoCr, cobalt-chromium-molybdenum alloy; SUS, stainless steel alloy.

**Table 3 materials-14-03702-t003:** Comparisons of frictional coefficient using Endurance (DePuy CMW, Blackpool, UK).

Sample	Surface Roughness Ra (μm)	95% CI	*p* Value *
CoCr vs. SUS	0.06	−0.11300–−0.01855	0.0073
CoCr vs. SUS	0.4	−0.06013–0.03436	0.5865
CoCr vs. Ti-15Mo-5Zr-3Al	0.1	0.22650–0.3209	<0.0001
CoCr vs. Ti-15Mo-5Zr-3Al	0.3	−0.14170–−0.04721	0.0002
CoCr vs. Ti-15Mo-5Zr-3Al	0.4	−0.09605–−0.00156	0.0432
CoCr vs. Ti-6Al-4V	0.1	0.23240–0.3269	<0.0001
CoCr vs. Ti-6Al-4V	0.4	0.01140–0.1059	0.016
SUS vs. Ti-15Mo-5Zr-3Al	0.4	−0.08316–0.01133	0.1332
SUS vs. Ti-6Al-4V	0.4	0.02428–0.1188	0.0037
Ti-15Mo-5Zr-3Al vs. Ti-6Al-4V	0.1	−0.04134–0.05315	0.8031
Ti-15Mo-5Zr-3Al vs. Ti-6Al-4V	0.2	−0.05122–0.04327	0.8665
Ti-15Mo-5Zr-3Al vs. Ti-6Al-4V	0.4	0.06020–0.1547	0.0073

* Generalised liner mixed effects model with samples as a random effect. CoCr, cobalt-chromium-molybdenum alloy; SUS, stainless steel alloy.

**Table 4 materials-14-03702-t004:** Comparisons of frictional coefficient using Palacos R (Heraeus Kulzer, Wehrheim, Germany).

Sample	Surface Roughness Ra (μm)	95% CI	*p* Value *
CoCr vs. SUS	0.06	−0.05416–0.03976	0.7596
CoCr vs. SUS	0.4	−0.04324–0.05067	0.8745
CoCr vs. Ti-15Mo-5Zr-3Al	0.1	0.17490–0.2689	<0.0001
CoCr vs. Ti-15Mo-5Zr-3Al	0.3	−0.16580–−0.0719	<0.0001
CoCr vs. Ti-15Mo-5Zr-3Al	0.4	−0.05236–−0.04156	0.8184
CoCr vs. Ti-6Al-4V	0.1	0.18640–0.2804	<0.0001
CoCr vs. Ti-6Al-4V	0.4	0.08748–0.1814	<0.0001
SUS vs. Ti-15Mo-5Zr-3Al	0.4	−0.05608–0.03784	0.6984
SUS vs. Ti-6Al-4V	0.4	0.08377–0.1777	<0.0001
Ti-15Mo-5Zr-3Al vs. Ti-6Al-4V	0.1	−0.03546–0.05846	0.6252
Ti-15Mo-5Zr-3Al vs. Ti-6Al-4V	0.2	−0.06714–0.02678	0.3925
Ti-15Mo-5Zr-3Al vs. Ti-6Al-4V	0.4	0.09289–0.1868	<0.0001

* Generalised liner mixed effects model with samples as a random effect. CoCr, cobalt-chromium-molybdenum alloy; SUS, stainless steel alloy.

**Table 5 materials-14-03702-t005:** Definitions of metal implant surface appearance and typical methods of manufacturing.

Surface Appearance	Typical Manufacturing Methods	Approximate Roughness Range Ra (μm)
Shiny	Polishing	0–0.1
Smooth	Machining, grinding, and mass finishing	0.1–0.4
Satin	Bead blasting and machining	0.4–1.0
Matte	Grit blasting combination grit + bead blasting	1.0–2.5
Rough	Aggressive grit blasting, plasma spraying, and sintering	2.5–12.5
Textured	Machining, casting, and forging	≥12.5

**Table 6 materials-14-03702-t006:** List of the major available cemented stems.

Stem Brand	Manufacturer	Design	Material	Surface Roughness (μm)
Charnley	DePuy International	Composite-beam	CoCr	0.75
Lubinas SP II	Waldemar Link	Composite-beam	CoCr	1.5
Exeter	Stryker	Taper-slip (double)	SUS	<0.05
C-stem	DePuy International	Taper-slip (triple)	SUS	0.02–0.17
CPT	Zimmer	Taper-slip (double)	CoCr	0.025–0.05

CoCr, cobalt-chromium-molybdenum alloy; SUS, stainless steel alloy.

## Data Availability

The data presented in this study are available on reasonable request from the corresponding author.

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
