# Peer review of "Relationship between the Surface Roughness of Material and Bone Cement: An Increased “Polished” Stem May Result in the Excessive Taper-Slip"

_materials, 2021, doi:10.3390/ma14133702_

Round 1

Reviewer 1 Report

This manuscript reports the study of stability of bone cement (Endurance and Palacos R) on different alloys (CoCr, SUS, Ti-6Al-4V, and Ti-15Mo-5Zr-3Al) in taper-slip stems for total hip arthroplasty (THA). Experiments measuring surface wettability and frictional coefficient were performed to evaluate effect of metal surface roughness on bone cement in order to prevent periprosthetic femoral fractures following THA.

To my opinion, data reported in this paper are complete, the manuscript is clearly written and the subject is appropriate for publications in Materials. My only objection concerns Figure 8: the authors have compared frictional coefficient calculated by the power approximation fitted curves for two values of surface roughness. In Figure 8 the behavior of different materials is very similar with no appreciable influence of surface roughness. Moreover, experimental values for titanium alloys samples at Ra = 0.06 mm do not exist in Table 2 and power approximation fitted curves reported in Figure 7 are different from those of other alloys. How the values reported in Figure 8 have been extrapoled?

Minor remark: a list of abbreviation would be useful.

Author Response

Comment (1): To my opinion, data reported in this paper are complete, the manuscript is clearly written and the subject is appropriate for publications in Materials. My only objection concerns Figure 8: the authors have compared frictional coefficient calculated by the power approximation fitted curves for two values of surface roughness. In Figure 8 the behavior of different materials is very similar with no appreciable influence of surface roughness. Moreover, experimental values for titanium alloys samples at Ra = 0.06 mm do not exist in Table 2 and power approximation fitted curves reported in Figure 7 are different from those of other alloys. How the values reported in Figure 8 have been extrapoled?

Response: Thank you for your question. Using a power approximation fitted curve in Figure 7, the expected frictional coefficient is calculated and a surface roughness of arbitrary Ra = 0.06 or 0.1 is selected. On page 7 line 232, we have changed the following sentence: “Using a power approximation fitted curve in Figure 7, the expected frictional coefficient is calculated. With a surface roughness of arbitrary Ra = 0.06 or 0.1 μm, the frictional coefficient of CoCr was less than that of SUS in both Endurance and Palacos R (Figure 8).”

Comment (2): Minor remark: a list of abbreviation would be useful.

Response: Thank you for your suggestion. As you have recommended, we have added a list of abbreviation.

Reviewer 2 Report

In this manuscript, Hirata et al study the relationship between the surface roughness of material and bone cement. Specifically the authors evaluated the surface wettability and frictional coefficient by using four different types of metal discs and two types of bone cement. This is a new angle to study the biomaterial effect contribute to periprosthetic femoral fractures. However, there are two minor concerns need to be addressed.

  1. The title of the manuscript need to be changed, since the study focus on evaluating the surface wettability and frictional coefficient and no data support “an excessively polished stem may cause periprosthetic frmoral fracture”.
  2. The samples size is small. The authors may need to make more metal discs with varying degrees of roughness, not only five.

Author Response

Comment (1): The title of the manuscript need to be changed, since the study focus on evaluating the surface wettability and frictional coefficient and no data support “an excessively polished stem may cause periprosthetic frmoral fracture”.

Response: Thank you for your suggestion. As you have recommended, we have changed the title of the manuscript to the following: “Relationship between the surface roughness of material and bone cement: an increased “polished” stem may result in the excessive taper-slip.”

Comment (Reviewer #2-2): The samples size is small. The authors may need to make more metal discs with varying degrees of roughness, not only five.

Response: Thank you for your suggestion. However, we assessed five metal discs and calculated a power approximation fitted curve in this study. On page 10 line 327, we have described the following sentence: “Some limitations of our study must be noted. First, the sample size was small. Although statistically significant data were acquired, the Ra parameters that could be compared were limited. Ideally, more samples should have been included.”

Round 2

Reviewer 2 Report

The author addressed all the concerns. I suggest to publish this manuscript.

This manuscript is a resubmission of an earlier submission. The following is a list of the peer review reports and author responses from that submission.